

# Source-Resolved Variability of Fine Particulate Matter and Human Exposure in an Urban Area

Pablo Garcia Rivera[1], Brian T. Dinkelacker[1], Ioannis Kioutsioukis[2],
Peter J. Adams[3,4], and Spyros N. Pandis[5,6]

[1]Department of Chemical Engineering, Carnegie Mellon University, Pittsburgh, PA, 15213

[2]Department of Physics, University of Patras, 26500, Patras, Greece

[3]Department of Civil and Environmental Engineering, Carnegie Mellon University, Pittsburgh, PA, 15213

[4]Department of Engineering and Public Policy, Carnegie Mellon University, Pittsburgh, PA, 15213

[5]Institute of Chemical Engineering Sciences (FORTH/ICE-HT), 26504, Patras, Greece

[6]Department of Chemical Engineering, University of Patras, 26500, Patras, Greece

## Abstract

Increasing the resolution of chemical transport model (CTM) predictions in urban areas is important to capture sharp spatial gradients in atmospheric pollutant concentrations and better inform air quality and emissions controls policies that protect public health. The chemical transport model PMCAMx was used to assess the impact of increasing model resolution on the ability to predict the source-resolved variability and population exposure to $PM_{2.5}$ at 36 x 36, 12 x 12, 4 x 4, and 1 x 1 km resolutions over the city of Pittsburgh during typical winter and summer periods (February and July 2017). At the coarse resolution, county-level differences can be observed, while increasing the resolution to 12 x 12 km resolves the urban-rural gradient. Increasing resolution to 4 x 4 km resolves large stationary sources such as power plants and the 1 x 1 km resolution reveals intra-urban variations and individual roadways within the simulation domain. Regional pollutants that exhibit low spatial variability such as $PM_{2.5}$ nitrate show modest changes when increasing the resolution beyond 12 x 12 km. Predominantly local pollutants such as elemental carbon





and organic aerosol have gradients that can only be resolved at the 1 x 1 km scale.
Contributions from some local sources are enhanced by weighting the average contribution
from each source by the population in each grid cell. The average population weighted
PM$_{2.5}$ concentration does not change significantly with resolution, suggesting that
extremely high resolution PM$_{2.5}$ predictions may not be necessary for effective urban
epidemiological analysis.

## 1. Introduction

Particulate matter with aerodynamic diameter less than 2.5 µm (PM$_{2.5}$) contributes

to poor air quality throughout large parts of the United States. These particles directly affect
visibility (Seinfeld and Pandis, 2006) and have been associated with long and short-term
health effects such as premature death due to cardiovascular disease, increased chance of
heart attacks and strokes, reduced lung development and function in children and people
with lung diseases such as asthma and increases in hospital admissions due to heart and
lung disease (Dockery and Pope, 1994).

At high resolutions, emissions from local sources such as commercial cooking, on-

road traffic, residential wood combustion, and industrial activities can have sharp gradients
that influence the geographical distribution of PM$_{2.5}$ concentrations. High-resolution
measurements of PM$_1$ have found gradients of up to ~2 µg m$^{-3}$ between urban background
sites and those with high local emissions (Gu et al., 2018; Robinson et al., 2018).

A key limiting factor on the modeling of particulate matter at high resolutions is

the geographical distribution of emissions. Previous studies have found that coarse grid
emissions interpolated to higher resolutions lead to small to modest improvements in model
predictive ability for ozone (Kumar and Russell, 1996; Arunachalam et al., 2006),
secondary organic aerosol (Stroud et al., 2011; Fountoukis et al., 2013) and nitrate
(Zakoura and Pandis, 2018; Zakoura and Pandis, 2019). Pan et al., (2017) used the default
approach from the U.S. Environmental Protection Agency (EPA) National Emissions
Inventory (NEI) to allocate county-based emissions to model grid cells at 4 x 4 and 1 x 1
km and found only small changes to model performance for NO$_x$ and O$_3$, while the 1 x 1
km case showed more detailed features of emissions and concentrations in heavily polluted
areas.



Improvements in the resolution of emission inventories have been focused on traffic
as this source exhibits significant variability at high resolutions. Recent approaches to
building high-resolution traffic inventories include origin-destination by vehicle class (Ma
et al., 2020), synthetic population mobility (Elessa Etuman and Coll, 2018) and fuel sales
combined with traffic counts (McDonald and McBride, 2014). Other sectors such as
biomass burning for residential heating and commercial cooking have been identified as
very uncertain in current inventories (Day et al., 2019). Recent versions of the NEI have
made progress addressing the total emissions and temporal distributions of biomass
burning and commercial cooking (Eyth and Vukovich, 2016), but there is still significant
uncertainty on their geographical location at a sub-county scale. Robinson et al. (2018)
found greatly elevated organic aerosol concentrations in the vicinity of numerous
individual restaurants and commercial districts containing groups of restaurants indicating
that commercial cooking is a source of large gradients on the urban scale.
Population density and socio-economic indicators of that population, such as
income or access to healthcare, show large gradients in the urban scale. It is important to
assess the exposure of different sub-populations to air pollutants and the resulting health
effects, a concept known as Environmental Justice (Anand, 2002).
We use the Particulate Matter Comprehensive Air quality Model with Extensions
(PMCAMx) to study the impact of increasing model resolution on the model's ability to
predict the variability, sources and population exposure of $PM_{2.5}$ concentrations on the
urban scale in Pittsburgh. We compare predicted variability at 36 x 36, 12 x 12, 4 x 4 and
1 x 1 km resolutions over the city of Pittsburgh during one typical summer and one typical
winter month of 2017. Additional sensitivity simulations were performed to determine
contributions from selected sources to concentrations. The results of the simulations are
used to estimate exposure to $PM_{2.5}$ at all resolutions and from the selected sources. A
detailed evaluation of the PMCAMx predictions against measurements will be the topic of
a future publication. Overall the model performance was similar to those in previous model
applications in the Eastern US (Fountoukis et al., 2013).


## 2. PMCAMx Description

The Particulate Matter Comprehensive Air quality Model with Extensions
(PMCAMx) (Tsimpidi et al., 2009; Karydis et al., 2010; Murphy and Pandis, 2010), uses
the framework of the CAMx model (Environ, 2006) to describe horizontal and vertical
advection and diffusion, emissions, wet and dry deposition, gas, aqueous and aerosol-phase
chemistry. A 10-size section aerosol sectional approach is used to dynamically track the
evolution of the aerosol mass distribution. The aerosol species modeled include sulfate,
nitrate, ammonium, sodium, chloride, elemental carbon, water, primary and secondary
organics, and other non-volatile aerosol components. The SAPRC (Statewide Air Pollution
Research Center) photochemical mechanism (Carter, 1999) is used for the simulation of
gas-phase chemistry. The version of SAPRC used here includes 237 reactions and 91
individual and surrogate species. For inorganic growth, a bulk equilibrium approach was
used, assuming equilibrium between the bulk inorganic aerosol and gas phases (Pandis et
al., 1993). The partition of the various semivolatile inorganic aerosol components and
aerosol water is determined using the ISORROPIA aerosol thermodynamics model (Nenes
et al., 1998). The primary and secondary organic aerosol components are described using
the volatility basis set approach (Donahue et al., 2006). For primary organic aerosol (POA)
ten volatility bins, with effective saturation concentrations ranging from $10^{-3}$ to $10^6$ µg
$m^{-3}$ at 298 K are used. Anthropogenic (aSOA) and biogenic (aSOA) are modeled with 4
volatility bins (1, 10, $10^2$, $10^3$ µg $m^{-3}$) (Murphy and Pandis, 2009) using $NO_x$ dependent
yields (Lane et al., 2008). More detailed descriptions of PMCAMx can be found in
Fountoukis et al. (2011) and Zakoura and Pandis (2018).

## 3. Model Application

PMCAMx was used to simulate air quality over the metropolitan area of Pittsburgh
during February and July 2017. For the base-case simulation we used a one-way nested
structure with a 36 x 36 km master grid covering the continental United States, with nested
grids of 12 x 12 km, 4 x 4 km in South Western Pennsylvania and a 1 x 1 km grid covering
the city of Pittsburgh, most of Allegheny County and the upper Ohio River valley (Figure
1a). The 1 x 1 km grid covers a 72 x 72 km area (Figure 1b).



The surface concentrations at the boundaries of the 36 x 36 km grid are shown in
Table S1 in the Supplementary Information. These values were applied to all upper air
layers assuming a constant mixing ratio. Horizontal wind components, vertical diffusivity,
temperature, pressure, water vapor, clouds, and rainfall were generated using the Weather
Research and Forecasting (WRF v3.6.1) model over the whole modeling domain with
horizontal resolution of 12 km. The data was interpolated to higher resolutions when
needed. Initial and boundary meteorological conditions for the WRF simulations were
generated from the ERA-Interim global climate re-analysis database, together with the
terrestrial data sets for terrain height, land-use, soil categories, etc. from the United States
Geological Survey (USGS) database. The WRF modeling system was prepared and
configured in a similar way as described by Gilliam and Pleim (2010). This configuration
is recommended for air quality simulations (Hogrefe et al., 2015; Rogers et al., 2013).
Emissions were calculated using the EPA's Emission Modeling Platform (v6.3) for
the National Emissions Inventory for 2011 (NEI11) (Eyth and Vukovich, 2016) using the
default 2017 projected values. Base emissions were calculated first at a 12 km resolution
for the full modeling domain using the Sparse Matrix Operator Kernel Emissions
(SMOKE) model and our WRF meteorological data. For the higher resolution grids, the
spatial surrogates provided with Platform v6.3 were used for all sectors except commercial
cooking and on-road traffic for which custom surrogates were developed. The emissions
by all sources together with the chemical composition are summarized in Table 1 (for the
winter period) and Table 2 (for the summer period)..
In this work, we used normalized restaurant count to distribute the commercial
cooking emissions in space in the 1x1 km inner domain. Geographical information was
collected for all locations labeled as "restaurant" from the freely accessible Google Places
Application Programming Interface (API) for the western Pennsylvania area, eastern Ohio
and northern West Virginia. Using this new spatial surrogate, $PM_{2.5}$ emissions from
commercial cooking are enhanced primarily in the Pittsburgh urban core with a maximum
increase of 1200 kg $g^{-1}$ $km^{-2}$ (Figure 2a).
To accurately capture spatial patterns of on-road traffic, we use the output of a link-
level, origin-destination by vehicle class traffic model of Pittsburgh (Ma et al., 2020). This
traffic model simulates traffic counts and speed by hour-of-day using observations from



Pennsylvania Department of Transportation sites throughout Pittsburgh. As expected,
emissions in areas with major highways are high (Figure 2b).

## 4.   PM$_{2.5}$ concentrations and sources during winter

### 4.1    Effect of grid resolution

The results of the simulations with the four resolutions for the winter period are
shown in Figure 3 and Figure 4. For the area of interest, the simulations at 36 x 36 km
resolves concentration fields at the county scale. The urban-rural gradient is resolved in the
12 x 12 km simulations. Increasing the resolution to 4 x 4 km, large stationary sources such
as power plants and large industrial installations are resolved. Finally, the resolution
increase to 1 x 1 km resolves the intra-urban variations in Pittsburgh and medium-sized
industrial installations.

In the winter period, the predicted maximum PM$_{2.5}$ concentration in the inner
domain increases from 10.4 µg m$^{-3}$ at 36x36 km, to 11.8 µg m$^{-3}$ at 12x12, to 12.9 µg m$^{-3}$ at
4x4, and finally to 16.4 µg m$^{-3}$ at 1x1 km (Figure 3), a 58% increase. On the other end, the
predicted minimum PM$_{2.5}$ concentration changes from 8.2 µg m$^{-3}$ at 36 x 36 km to 7 µg
m$^{-3}$ at 12 x 12 and remains practically the same at even higher resolutions. This corresponds
to the "background" concentration level for the area during the simulation period, so further
resolution enhancements do not change this value. The standard deviation of the predicted
concentration can be used as a measure of the concentration variability in the area. This
standard deviation changes from 0.9 µg m$^{-3}$ at 36x36, to 1.24 µg m$^{-3}$ at 12x12, to 1.45 µg
m$^{-3}$ at 4x4 and to 1.35 µg m$^{-3}$ at 1x1 km. These results indicate an increase of the PM$_{2.5}$
variability by 50% when one moves from the coarse to the finest resolution. However, most
of this change in variability (38% out of the 50%) appears when one moves from 36x36 to
12x12 km.

Elemental carbon is a primary aerosol component with sources that are quite
variable in space. In winter, the predicted maximum PM$_{2.5}$ EC increased by a factor of 2.9,
from 0.6 µg m$^{-3}$ at the 36 x 36 km resolution to 1.6 µg m$^{-3}$ at 1 x 1 km (Figure 3). The
predicted maximum EC is, as expected in the Pittsburgh downtown area. On the other hand,
the predicted minimum of EC is reduced by only 0.1 µg m$^{-3}$, from 0.34 µg m$^{-3}$ at 36x36
km to 0.24 µg m$^{-3}$ at resolutions lower or equal than 4x4 km. The standard deviation of the





predicted EC almost doubles from 0.1 µg m$^{-3}$ at 36 x 36 km to 0.18 µg m$^{-3}$ at 1 x 1 km.
Approximately 50% of this increase in variability appears in the transition from the coarse
to the intermediate resolution of 12 x 12 km. The fine and the finest resolutions are needed
to resolve the other half of the predicted variability.

During this winter period a significant fraction (79%) of the OA in the Pittsburgh

area is primary and therefore the higher resolution results in increases of the predicted
maximum concentrations in space from 2.8 µg m$^{-3}$ at the coarse resolution to 3.7 µg m$^{-3}$ at
the intermediate to 4.8 µg m$^{-3}$ at the finest resolution (Figure 3). This corresponds to an
increase by a factor of 1.7, more than the change for total PM$_{2.5}$, but much less than that
for EC. The predicted maximum is located in downtown Pittsburgh, with additional
hotspots in neighboring counties that are resolved at the fine and finest resolution. The
predicted minimum changes from 2.1 µg m$^{-3}$ at 36x36 to 1.7 µg m$^{-3}$ at 12x12 with small
reductions at higher resolutions. The variability (standard deviation) of the OA
concentration field of the predicted concentration increases by a factor of approximately
1.6 from 0.35 µg m$^{-3}$ at 36 x 36, to 0.51 µg m$^{-3}$ at 12 x 12 km. The increase is small at even
higher resolutions with the standard deviation of OA reaching 0.53 µg m$^{-3}$ at 1 x 1 km (an
increase by a factor of 1.7).

The predicted fine nitrate levels are relatively high ranging from 1.78 to 2.24 µg

m$^{-3}$ in the coarse-resolution simulation. This is expected in this wintertime period due to
the partitioning of nitric acid and ammonium in the particulate phase. This predicted
concentration range increases to 1.5-2.24 µg m$^{-3}$ in the finest scale simulation with higher
levels in the northeast of the domain. The standard deviation of the predicted concentration
does not show any significant trend changing from 0.19 µg m$^{-3}$ at 36 x 36 to 0.15 µg m$^{-3}$
at 1 x 1 km.

For PM$_{2.5}$ ammonium, changes with increasing resolution are modest with the

predicted minimum being reduced from 1.07 µg m$^{-3}$ at 36x36 to approximately 0.95 µg
m$^{-3}$ at all other higher resolutions. The predicted maximum stays relatively constant
between 1.25 µg m$^{-3}$ and 1.27 µg m$^{-3}$ at all resolutions. As with nitrate, the standard
deviation does not show any significant trend changing from 0.08 µg m$^{-3}$ at 36 x 36, to 0.09
µg m$^{-3}$ at 12 x 12, to 0.07 µg m$^{-3}$ at 4 x 4 and 1 x 1 km resolutions.





## 4.2 Source Apportionment


We performed zero-out simulations in the 1x1 km Pittsburgh grid to determine the
local contributions of eight source categories to the total $PM_{2.5}$. The local sources
quantified included: commercial cooking, industrial, biomass burning, on-road traffic,
power generation, and miscellaneous area sources. The miscellaneous area sources sector
includes a large variety of emission sources that are not classified in any of the sources in
Table 2. These include chemical manufacturing, solvent utilization for surface coatings,
degreasing and dry cleaning, storage and transport of petroleum products, waste disposal
and incineration, and cremation. The emissions from agricultural dust, river barges, off-
road equipment, oil-gas activities, and rail were grouped on the "others" source. All
emissions (particulate and gas-phase) from each source were set to zero, and the results of
the zero-out simulation were subtracted from those of the baseline simulation to estimate
the corresponding source contribution. The contribution of long-range transport from
outside the inner domain was also estimated by setting all local sources to zero.
Biomass burning is used during the winter for residential heating and recreation.
This source contributes a maximum of 3.31 µg m$^{-3}$ in Cranberry, a northern suburb of
Pittsburgh located in the neighboring Butler county. In the downtown Pittsburgh area, the
contribution from biomass burning accounts for 7% of the $PM_{2.5}$. This source shows the
highest variability with a standard deviation of 0.5 µg m$^{-3}$.
The maximum contribution of 8.05 µg m$^{-3}$ from industry is predicted near a cluster
of industrial facilities in the town of Butler, 37 km northwest of Pittsburgh. The maximum
$PM_{2.5}$ concentration of the modeling domain is located here. In this location long-range
transport contributes 37% of the $PM_{2.5}$ followed by industrial sources with 49% and
biomass burning with 7%. On average, the contribution from industrial sources is low with
3.7%. In downtown Pittsburgh, the contribution is lower still with 2%.
On-road traffic emissions are most important in major highway intersections and
river crossings surrounding downtown Pittsburgh with a maximum contribution of 3.9 µg
m$^{-3}$ accounting for 24% of the $PM_{2.5}$ in this area. On average, on-road traffic contributes
2.5% of the $PM_{2.5}$ mass. The contribution from on-road traffic shows higher variability
(standard deviation: 0.36 µg m$^{-3}$) since this sector contributes significantly to areas adjacent
to the network of highways that radiates from the Pittsburgh downtown.





On average, commercial cooking emissions contribute 0.7% of the $PM_{2.5}$ in the

modeling domain with a maximum contribution of 2.44 µg m$^{-3}$ in downtown Pittsburgh,
with smaller contributions in the surrounding urban area. Cooking is predicted to account
for 16% of the $PM_{2.5}$ mass in downtown Pittsburgh. The contribution from commercial
cooking is localized around downtown Pittsburgh and therefore shows little variability
throughout the domain with a standard deviation of 0.1 µg m$^{-3}$.

The miscellaneous area source sector contributes 6% of the $PM_{2.5}$ on average. Since

this sector encompasses a variety of sources and activities, its contribution shows
significant variability with a standard deviation of 0.34 µg m$^{-3}$.The maximum contribution
is located in the Pittsburgh urban core with 1.64 µg m$^{-3}$, accounting for 11% of the $PM_{2.5}$.

The power generation sector contributes a maximum of 0.63 µg m$^{-3}$ in the plume

of the Bruce Mansfield power plant northwest of Pittsburgh. This sector shows the smallest
variability with 0.09 µg m$^{-3}$.

Long-range transport from outside the inner modeling domain is the major source

of $PM_{2.5}$ during this period contributing an average of 74%. This contribution varies from
7.1 µg m$^{-3}$ in the southeast corner of the domain decreasing in the direction of the Pittsburgh
urban core where the contribution is reduced to 5.9 µg m$^{-3}$. In areas where there are
significant local emissions such as the Pittsburgh downtown, the contribution from long-
range transport decreases to 38%.

Contributions for all remaining sources are largest in the Pittsburgh downtown with

0.74 µg m$^{-3}$, accounting for 5% of the $PM_{2.5}$. This sector also significantly contributes on
the Ohio and Monongahela river valleys, where there is important rail and river traffic. On
average, these sources contribute 3% of the $PM_{2.5}$ and show a moderate variability with a
standard deviation of 0.1 µg m$^{-3}$.

For all local sources, the minimum contribution is close to zero (less than 0.1 µg

m$^{-3}$) and is located at the southwestern corner of the domain, near the Ohio – West Virginia
border.





## 5. PM$_{2.5}$ concentrations and sources during summer

### 5.1 Effect of grid resolution

The predicted PM$_{2.5}$ concentrations in the simulated summer period are lower than during the winter period and more uniform, however, the qualitative behavior of the model at the different scales remains the same (Figure 6). The standard deviation of the PM$_{2.5}$ increases from 0.28 µg m$^{-3}$ at 36 x 36, to 0.57 µg m$^{-3}$ at 12 x 12, to 0.72 µg m$^{-3}$ at 4 x 4 and to 0.82 µg m$^{-3}$ at 1 x 1 km. At the finest scale, the predicted variability in the summer is 61% of that in the winter. Similar to the winter period, the predicted maximum PM$_{2.5}$ concentration changes significantly with increasing resolution. The predicted maximum PM$_{2.5}$ increases from 6.4 µg m$^{-3}$ at the coarse to 15.3 µg m$^{-3}$ at the fine resolution. The finest scale better resolves the concentration field in the cluster of industrial installations 37 km northwest of Pittsburgh. The minimum PM$_{2.5}$ drops from 6.5 µg m$^{-3}$ at 36 x 36 to 5.3 µg m$^{-3}$ at 12 x 12, and then to 4.7 µg m$^{-3}$ at 1 x 1 km. As in the winter period, the moderate resolution appears to capture the majority of the concentration change from increasing resolution (67%).

The average EC is lower during the summer with 0.28 µg m$^{-3}$ versus 0.43 µg m$^{-3}$ in the winter. The standard deviation of the predicted average EC increases from 0.06 µg m$^{-3}$ at 36 x 36, to 0.09 µg m$^{-3}$ at 12 x 12, to 0.11 µg m$^{-3}$ at 4 x 4 km, and to 0.13 µg m$^{-3}$ at 1 x 1 km. The peak average EC is located in downtown Pittsburgh and increases by a factor of 3.6 (from 0.35 to 1.27 µg m$^{-3}$) moving from the coarse to the finest resolution. It is noteworthy that the peak is 38% less than that of the winter when the coarse resolution is used, but only 21% when the finest resolution is used. The concentration range (difference between the maximum and the minimum) increases from 0.13 µg m$^{-3}$ to 1.12 µg m$^{-3}$ moving from the coarse to the finest resolution. This increase by a factor of 8.6 shows the importance of the local variations of a primary species like EC in an urban area in both summer and winter.

The OA concentration field is quite uniform at the coarse-scale varying by only 0.17 µg m$^{-3}$ (from 1.72 to 1.89 µg m$^{-3}$) with a standard deviation of 0.07 µg m$^{-3}$ (Figure 6). Variablility increases significantly when one moves to the finest scale, with the range increasing to 2.24 µg m$^{-3}$ (from 1.55 to 3.79 µg m$^{-3}$) and the standard deviation of the OA



field increases to 0.2 µg m$^{-3}$. The use of the finest scale appears to be needed for the
resolution of the OA high concentration areas in the summer more than in the winter.
The PM$_{2.5}$ sulfate levels during the summer period are on average 12% higher
during the summertime period. At the coarse and intermediate scales, the predicted average
concentration fields have relatively little structure (Figure 7). The corresponding
concentration ranges are relatively narrow (0.05 µg m$^{-3}$ at 36 x 36 km and 0.42 µg m$^{-3}$ at
12x12 km). However, a different picture emerges at the fine and especially the finest scales.
The plumes from the major power plants can be clearly seen at these higher resolutions.
The maximum increased by 0.5 µg m$^{-3}$ from the coarse scale to the finest scale while the
minimum is reduced from 1.78 µg m$^{-3}$ at 36 x 36 to 1.05 µg m$^{-3}$ at 12 x 12, to 0.95 µg m$^{-3}$
at 4 x 4 and 1 x 1 km.  The standard deviation of the predicted sulfate concentration field
at the coarse resolution is low and similar to that in winter, 0.02 µg m$^{-3}$. However, the
variability at the finest scale in the summer (0.13 µg m$^{-3}$ at 1x1 km) is twice the predicted
variability in the winter.
The predicted summertime nitrate concentrations are quite low in the area (average
0.5 µg m$^{-3}$ in the coarse and 0.46 µg m$^{-3}$ in the finest resolution). The predicted minimum
decreases from 0.42 µg m$^{-3}$ at 36 x 36 to 0.39 µg m$^{-3}$ at 12 x 12, to 0.34 µg m$^{-3}$ at 4 x 4, and
to 0.3 µg m$^{-3}$ at 1 x 1 km. The predicted maximum concentration increases from 0.56 µg
m$^{-3}$ at the coarse scale to 0.71 µg m$^{-3}$ at the intermediate scale and stays relatively constant
at higher resolutions. The concentration field is quite uniform with a standard deviation
ranging from 0.06 to 0.09 µg m$^{-3}$ for all scales. However, due to the reduction in the
predicted minimum the concentration range increases from 0.14 µg m$^{-3}$ at the coarse
resolution to 0.37 µg m$^{-3}$ at the finest resolution.
The PM$_{2.5}$ ammonium concentration field is quite uniform at all resolutions (Figure
7). The concentration range increases from 0.04 to 0.22 µg m$^{-3}$ moving from the coarse to
the finest resolution and the standard deviation increases from 0.02 to 0.04 µg m$^{-3}$.

**5.2    Source Apportionment**
During summer, residential biomass burning is minimal. This source contributes a
maximum of 0.04 µg m$^{-3}$ and an average of 0.007 µg m$^{-3}$,accounting for 0.6% of the
average total PM$_{2.5}$.


Power generation sources have the highest average contribution to total $PM_{2.5}$ of
all the local sources of 10%. Industrial sources account for 6% of the average $PM_{2.5}$ but are
the most important contributor in the point of the modeling domain with the maximum
predicted $PM_{2.5}$ conentration. At this location in Butler County, industrial sources account
for 58% of total $PM_{2.5}$
As in the winter period, on-road traffic emissions have the largest contribution to
the $PM_{2.5}$ in the downtown Pittsburgh area where four large highways intersect. In this
location on-road traffic contributes 26% of the $PM_{2.5}$. On average, local on-road traffic
contributes around 3% of the $PM_{2.5}$ mass. During the summer period, the variability of the
on-road traffic contribution is slightly lower with 0.33 µg m$^{-3}$ compared with 0.36 µg m$^{-3}$
during winter.
Commercial cooking emissions contribute a maximum of 2.08 µg m$^{-3}$ to the
average total $PM_{2.5}$ in downtown Pittsburgh. This source accounts for 17% of the $PM_{2.5}$ in
the city but only 1% for the entire modeling domain.
On average, the miscellaneous area sources sector contributes 0.26 µg m$^{-3}$
accounting for 4.3% of the $PM_{2.5}$. In downtown Pittsburgh, where the contribution is
highest, this source contributes 7% of the $PM_{2.5}$.
Unlike in the winter period, the plumes from major powerplants in the Ohio river
valley are clearly resolved in the summer. The power generation sector contributes a
maximum of 2.4 µg m$^{-3}$ in the plume of the Bruce Mansfield power plant northwest of
Pittsburgh. On average, the 9.4% contribution from this sector to the $PM_{2.5}$ is much larger
than in the winter where it only contributed 2.3%. The plume from the Mitchell power
plant in the southwest corner of the modeling domain is clearly resolved and reaches all
the way to the city. This increases the contribution from power generation to the $PM_{2.5}$ in
the downtown core from 0.22 µg m$^{-3}$ in the winter to 0.61 µg m$^{-3}$ in the summer. The
maximum contribution of 8.98 µg m$^{-3}$ from industrial sources is a cluster of industrial
facilities in the town of Butler, northwest of Pittsburgh.
Long-range transport from sources outside the region contributes a maximum of
5.2 µg m$^{-3}$ in the southeast corner of the domain decreasing in the direction of the Pittsburgh
northern suburbs where the contribution is minimal with 4.1 µg m$^{-3}$. On average, long-





range transport accounts for 72% of the PM$_{2.5}$ mass. In downtown Pittsburgh, long-range
transport contributes 4.24 µg m$^{-3}$ accounting for 35% of the PM$_{2.5}$.
On average, the contribution from all remaining sources is 3.6% and shows a
moderate variability of 0.10 µg m$^{-3}$. The contribution from these sources is maximal in
downtown Pittsburgh with 0.78 µg m$^{-3}$ accounting for 6% of the PM$_{2.5}$.
For all local sources, the minimum contribution is close to zero (less than 0.1 µg
m$^{-3}$) and is located at the northwestern corner of the domain, near the Ohio – Pennsylvania
border.
Relative contributions of all local sources to domain average predicted total PM$_{2.5}$
are shown in Figure 9. The largest differences between February and July are the
contributions from biomass burning and power generation. In the winter, biomass burning
is the most important local source of PM$_{2.5}$, contributing over 8%. In the summer, this
source contributes much less than 1% to total PM$_{2.5}$. This discrepancy can easily be
explained by the lack of residential wood combustion in the warmer months of the year.
Power generation is a significantly more important source in July than in February. This is
likely a result of a lower mixing height in the winter combined with emissions plumes from
power plants in the Ohio river vally originating from very tall stacks.
The relative contributions of local sources to average predicted total PM$_{2.5}$ in the
maximum concentration cell in Butler County and in downtown Pittsburgh are shown in
Figures 10 and 11, respectively. The dominant local source in the Butler County location
is industrial emissions, due to the proximity of various industrial installations in this area.
Industrial sources here account for around 49% of total PM$_{2.5}$ in February and 58% of total
PM$_{2.5}$ in July. A lot of the difference in industrial PM$_{2.5}$ at the Butler County location
between months is made up by biomass burning in February, which accounts for 7% more
of the total compared to July.

**6. Exposure to PM$_{2.5}$**
The population data in the inner domain from the 2010 U.S. census was used to
estimate the exposure of the population in the Pittsburgh area to model predictions of PM$_{2.5}$
during winter of 2017 at the different grid resolutions. We ranked the average PM$_{2.5}$
concentrations from all the cells in the modeling domain and created bins of 0.2 µg m$^{-3}$. A





sum of the population from all the grid cells that fall within each concentration bin was
calculated and divided by the total population of the inner grid to construct population
exposure histograms.

**6.1      Winter PM$_{2.5}$ Exposure**

Figure 12 shows the population exposure histograms for the Pittsburgh area (inner

domain) for each model resolution. At the coarse resolution, there are only four PM$_{2.5}$
values and 46% of the population is exposed to a concentration of 10.4 µg m$^{-3}$ with
decreasing exposure with PM$_{2.5}$ concentration. At a 12 km resolution, the low
concentration side of the distribution is better resolved but gaps can still be observed at
higher levels. At this intermediate resolution, the largest fraction of the population (15%)
is exposed to PM$_{2.5}$ concentrations of 11.8 µg m$^{-3}$.

When the resolution is increased to 4 km the biggest improvements on the model

ability to resolve the exposure distribution happen at concentrations higher than 9.4 µg
m$^{-3}$. At the fine resolution, no gaps appear in the distribution. A maximum of 12% of the
population is exposed to PM$_{2.5}$ concentrations of 12 µg m$^{-3}$ while at the highest
concentration of 12.8 µg m$^{-3}$ 3% are exposed. At the 1 km resolution, the distribution is
much smoother due to the ability of this finest grid to capture local gradients. The largest
fraction of the population (6%) is exposed to PM$_{2.5}$ concentrations of 9.2 µg m$^{-3}$. At the
highest concentration of 14.4 µg m$^{-3}$ the exposed population is less than 0.1% as this
maximum point is located near industrial installations 37 km northwest of Pittsburgh where
the population density is very low.

At resolutions of 36 km, 12 km, 4 km, and 1 km the predicted average population

weighted total PM$_{2.5}$ concentration during February 2017 is 9.74 µg m$^{-3}$, 9.77 µg m$^{-3}$, 10.28
µg m$^{-3}$, and 10.00 µg m$^{-3}$, respectively. This represents an increase of only 2.6% when
moving from lowest to highest resolution. Relative contributions of local sources to
average population weighted PM$_{2.5}$ concentration is shown in Figure 14. Compared to the
domain average PM$_{2.5}$ concentrations (Figure 9), many local source contributions are
enhanced in terms of average population exposure. In February, weighting PM$_{2.5}$
concentrations by population increases the contribution from biomass burning from 8.3%
to 11.7%. Other notable increases include onroad traffic (2.5% to 6.5%), and miscellaneous





area sources (5.9% to 9.2%). Other local source contributions to population weighted PM$_{2.5}$
were similar to the corresponding non-weighted concentrations.

**6.2    Summer PM$_{2.5}$ Exposure**
Figure 14 shows the population exposure for each simulation grid during the
summer period. At the coarse resolution, 88% of the population is exposed to a
concentration of 7 to 7.2 µg m$^{-3}$. At 12 x 12 km resolution, the exposure distribution is
better resolved but a gap is still present at 7.2 µg m$^{-3}$ and exposure to PM$_{2.5}$ concentrations
above 7.6 µg m$^{-3}$ is not resolved at all. At this intermediate resolution, the largest fraction
of the population (19%) is exposed to PM$_{2.5}$ concentrations of 7.4 µg m$^{-3}$. Increasing the
resolution to 4 x 4 km both shifts the distribution to slightly lower concentrations and
resolves exposure to higher PM$_{2.5}$ concentrations than with the 12 x 12 km grid. At this
resolution, 14% of the population is exposed to 6.4 µg m$^{-3}$ and smaller portions of the
population are exposed to concentrations higher than 8.0 µg m$^{-3}$. Moving to the highest
resolution grid further resolves the exposure distribution. Most notably, 1 x 1 km resolution
reveals a bimodal distribution of population exposure, with one peak centered around 6.0
µg m$^{-3}$ and another centered around 7.4 µg m$^{-3}$. This likely corresponds to one subset of
the population in the urban areas of Pittsburgh who are exposed to higher PM$_{2.5}$
concentrations and another subset representing the surrounding suburban areas.
At resolutions of 36 km, 12 km, 4 km, and 1 km the predicted average population
weighted total PM$_{2.5}$ concentration during February 2017 is 7.06 µg m$^{-3}$, 6.78 µg m$^{-3}$, 7.00
µg m$^{-3}$, and 6.99 µg m$^{-3}$, respectively. This represents just a 1% decrease between the
lowest and highest resolutions. Similar to the effect seen in February, weighting PM$_{2.5}$
concentrations by population increases the contribution from onroad traffic from 3.3% to
8.9% in July. Contributions from miscellaneous area sources also increased (4.3% to 7.1%)
when weighting by population. The population weighted contribution from power
generation sources in July decreased from the non-weighted value from 9.4% to 8.3%. All
other local source contributions to population weighted PM$_{2.5}$ in July were similar to the
non-weighted values.



## 7. Conclusions


We applied the PMCAMx chemical transport model over the city of Pittsburgh for
the simulation periods of February and July 2017 using a series of telescoping grids at 36
x 36 km, 12 x 12 km, 4 x 4 km and 1 x 1 km. Emissions were calculated using 2017
projections from the 2011 NEI. Emissions were distributed geographically using the spatial
surrogates provided with the NEI11 for all grids. For commercial cooking, a new 1 x 1 km
spatial surrogate was developed using restaurant count data from the Google Places API.
Traffic model data was used to develop a 1 x 1 km spatial surrogate for on-road traffic
emissions.
At the coarse resolution, county-level differences can be observed. Increasing the
resolution to 12 x 12 km resolves the urban-rural gradient and further increasing to 4 x 4
resolves large stationary sources such as power plants. Only at the finest resolution intra-
urban variations and individual roadways are resolved. Low variability, regional pollutants
such as nitrate show limited improvement after increasing the resolution to 12 x 12 km
while predominantly local pollutants such as elemental carbon and winter organic aerosol
have gradients that can only be resolved at the finest resolution.
Biomass burning shows the largest variability during the winter period with many
local maxima and significant emissions within the city and in the suburbs. During the
summer contributions from this source are negligible. In contrast with the winter period,
during the summer the plumes from large power plants in the Ohio river valley can be
resolved. These plumes are rich in sulfates and start being resolved at 4 x 4 km with
significant detail added at 1 x 1 km. During both periods the largest contributing source to
the average $PM_{2.5}$ is particles from outside the modeling domain.
The ability of the model to resolve the exposure distribution increases at different
rates according to the concentration. A significant improvement in resolving exposure to
concentrations below 9.4 µg m$^{-3}$ in the winter and below 7.0 µg m$^{-3}$ in the summer is
achieved by increasing the resolution to 12 x 12 km. Only at the finest resolution is the
exposure to concentrations above 9.6 µg m$^{-3}$ in the winter and above 8.6 µg m$^{-3}$ in the
summer fully resolved as well as the impact of high concentration spots.
The average exposure in terms of average contribution to population weighted
$PM_{2.5}$ concentrations of some local sources is enhanced compared to the non-weighted



average PM$_{2.5}$ concentrations. In February, weighting by population enhanced the
contributions from biomass burning, onroad traffic, and miscellaneous area sources by 3-
4%. In July, the contributions from onroad traffic and miscellaneous area sources also
increased by 3-5% from this procedure.
It was determined that increasing simulation grid resolution from 36 x 36 km to 1
x 1 km had minimal effect on the predicted domain average population weighted PM$_{2.5}$
concentration. Moving from the lowest to highest grid resolution increased the predicted
average population weighted PM$_{2.5}$ by less than 3%. In July, the average decreased by 1%.
This negligible change in the predicted average exposure to PM$_{2.5}$ suggests that extremely
high resolution predictions of urban PM$_{2.5}$ pollution may not be necessary for accurate
epidemiological analysis, however the increased neighborhood scale resolution could be
vital for topics related to environmental justice

**8. Code and data availability**

The    code    and    simulation    results    are    available    upon    request
(spyros@chemeng.upatras.gr).

**9. Supplement**


**10. Author contributions**

P.G.R. and B.T.D. performed the PMCAMx simulations, analyzed the results and
wrote the manuscript. P.G.R. prepared the anthropogenic emissions and other inputs for
the simulations. I.K. set-up the WRF simulations and assisted in the preparation of the
meteorological inputs. S.N.P. and P.J.A. designed and coordinated the study and helped
in the writing of the paper. All authors reviewed and commented on the manuscript.

**11. Competing interests**

The authors declare that they have no conflict of interest.




## 12. Financial Support


This work was supported by the Center for Air, Climate, and Energy Solutions
(CACES) which was supported under Assistance Agreement No. R835873 awarded by the
U.S. Environmental Protection Agency and the Horizon-2020 Project REMEDIA of the
European Union under grant agreement No 874753.






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




**Table 1.** PM$_{2.5}$ emissions by source for the 1 x 1 km Pittsburgh domain (February 2017).

| Source Type | Emissions (kg d$^{-1}$ km$^{-2}$) | | | | | | | | |
|---|---|---|---|---|---|---|---|---|---|
| | PM$_{2.5}$ | OA | EC | Chl. | Na | Amm. | Nitrate | Sulfate | Other |
| Agricultural dust | 68.7 | 9.7 | 0.4 | 0.2 | 0.1 | 0.1 | 0.1 | 0.7 | 57.2 |
| River barges | 19.0 | 4.2 | 14.7 | 0.0 | 0.0 | 0.0 | 0.0 | 0.1 | 0.1 |
| Cooking | 242 | 223 | 8.3 | 2.2 | 0.8 | 0.0 | 1.1 | 0.6 | 6.0 |
| Misc. area sources | 683 | 445 | 56.7 | 30.5 | 3.0 | 5.6 | 1.7 | 42 | 97.8 |
| Off-road | 147 | 56.2 | 73.1 | 0.3 | 0.1 | 0.0 | 0.3 | 1.1 | 16.1 |
| Oil-gas (Area) | 35.3 | 1.7 | 0.0 | 0.0 | 0.0 | 0.0 | 0.1 | 8.3 | 23.2 |
| On-road traffic | 188 | 84.6 | 75.2 | 0.3 | 0.1 | 1.8 | 0.6 | 8.3 | 16.4 |
| Rail | 40.7 | 8.9 | 31.4 | 0.0 | 0.0 | 0.0 | 0.0 | 0.1 | 0.2 |
| Biomass burning | 1,869 | 1,696 | 105 | 5.6 | 1.8 | 2.8 | 3.6 | 7.7 | 46.3 |
| Power generation | 3,517 | 201 | 194 | 2.8 | 0.0 | 15.7 | 2.6 | 460 | 2,641 |
| Industrial | 1,106 | 192 | 134 | 79.4 | 65.3 | 10.1 | 21.1 | 173 | 428 |
| Oil-gas (point) | 2.8 | 1.0 | 1.1 | 0.0 | 0.0 | 0.0 | 0.1 | 0.2 | 0.5 |


**Table 2.** PM$_{2.5}$ emissions by source for the 1 x 1 km Pittsburgh domain (July 2017).

| Source Type | Emissions (kg d$^{-1}$ km$^{-2}$) | | | | | | | | |
|---|---|---|---|---|---|---|---|---|---|
| | PM$_{2.5}$ | OA | EC | Chl. | Na | Amm. | Nitrate | Sulfate | Other |
| Agricultural dust | 67.3 | 8.9 | 0.4 | 0.1 | 0.1 | 0.1 | 0.1 | 0.7 | 56.9 |
| River barges | 19.0 | 4.2 | 14.7 | 0.0 | 0.0 | 0.0 | 0.0 | 0.1 | 0.1 |
| Cooking | 242 | 223 | 8.3 | 2.2 | 0.8 | 0.0 | 1.1 | 0.6 | 6 |
| Misc. area sources | 593 | 392 | 49.1 | 28.5 | 2.5 | 5.3 | 1.1 | 33 | 81.6 |
| Off-road | 205 | 83.5 | 92.9 | 0.2 | 0.1 | 0.0 | 0.4 | 1.1 | 27.3 |
| Oil-gas (Area) | 35.9 | 1.9 | 0.0 | 0.0 | 0.0 | 0.0 | 0.1 | 8.9 | 25.0 |
| On-road traffic | 162 | 67.6 | 66 | 0.4 | 0.1 | 1.5 | 0.5 | 8.6 | 17.2 |
| Rail | 40.7 | 8.9 | 31.4 | 0.0 | 0.0 | 0.0 | 0.0 | 0.1 | 0.2 |
| Biomass burning | 24.3 | 22 | 1.4 | 0.0 | 0.0 | 0.0 | 0.0 | 0.1 | 0.6 |
| Power generation | 3,780 | 216 | 208 | 3.1 | 0.0 | 16.9 | 2.8 | 495 | 2,840 |
| Industrial | 1,050 | 188 | 133 | 67.3 | 56.2 | 9.9 | 21.0 | 165 | 412 |
| Oil-gas (point) | 2.8 | 1.0 | 1.1 | 0.0 | 0.0 | 0.0 | 0.1 | 0.2 | 0.5 |







**Table S1**. Outer (CONUS) boundary condition concentrations of major aerosol species.

| Component | Concentration ($\mu g\ m^{-3}$) | | | |
|---|---|---|---|---|
| | West | East | South | North |
| Nitrate | 0.01 | 0.01 | 0.03 | 0.03 |
| Ammonium | 0.14 | 0.25 | 0.24 | 0.16 |
| Sulfate | 0.64 | 1.12 | 0.81 | 0.68 |
| Elemental carbon | 0.04 | 0.05 | 0.09 | 0.03 |
| Organic aerosol (Winter) | 0.20 | 0.16 | 0.58 | 0.80 |
| Organic aerosol (Summer) | 0.80 | 0.80 | 0.80 | 0.80 |









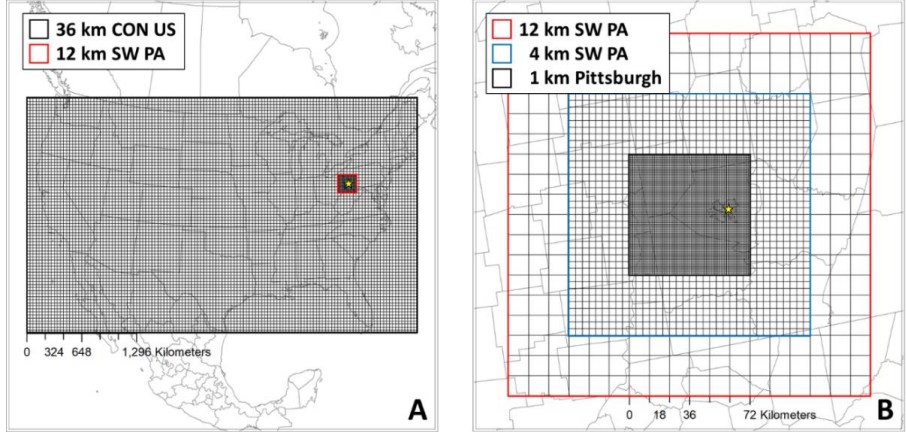


**Figure 1.** Modeling domain used for the PMCAMx simulations. (**A**) 36 x 36 km continental U.S. grid. (**B**) 12 x 12 and 4 x 4 km South Western Pennsylvania grids, and 1 x 1 km Pittsburgh nested grids.



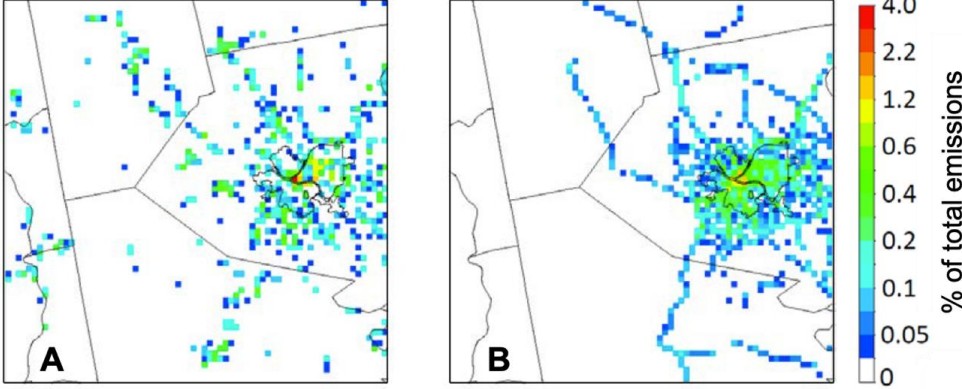


**Figure 2.** Percentage of $PM_{2.5}$ emissions in each 1x1 km computational cell for: (**A**) commercial cooking and (**B**) on road traffic.






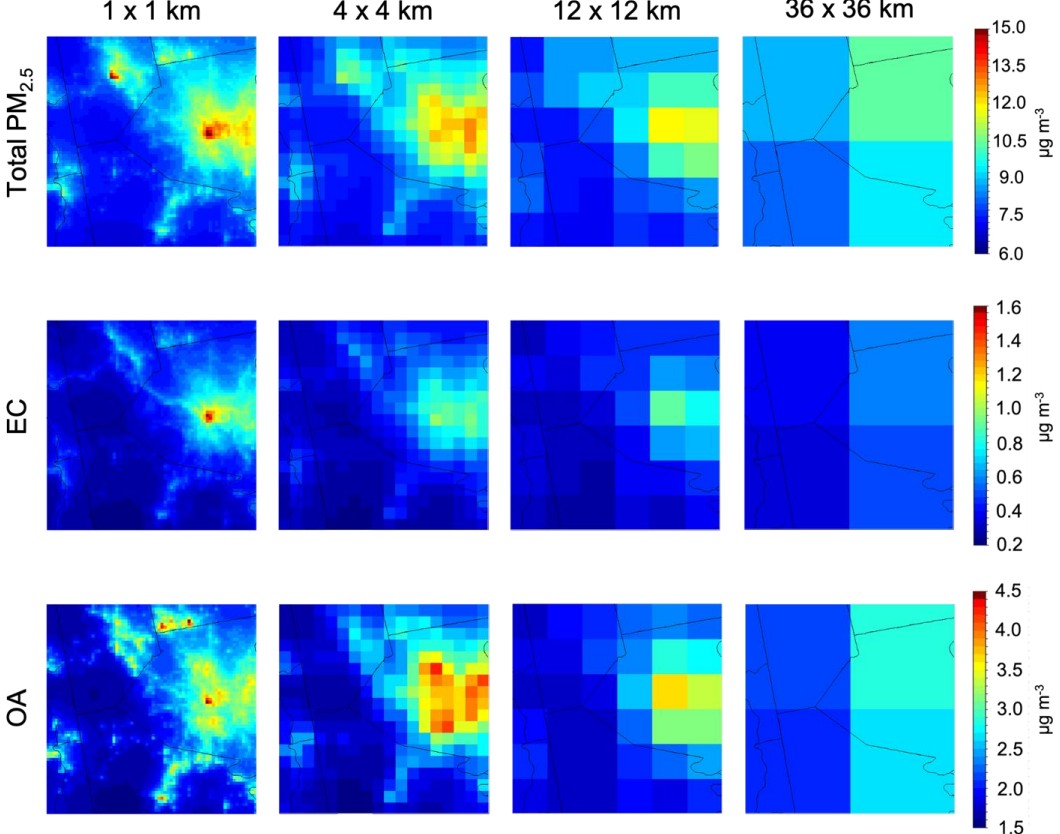


**Figure 3** Average predicted ground-level concentration of total PM$_{2.5}$, EC, and OA at 36 x 36, 12 x 12, 4 x 4 and 1 x 1 km resolutions during February 2017. Different scales are used for the various maps.








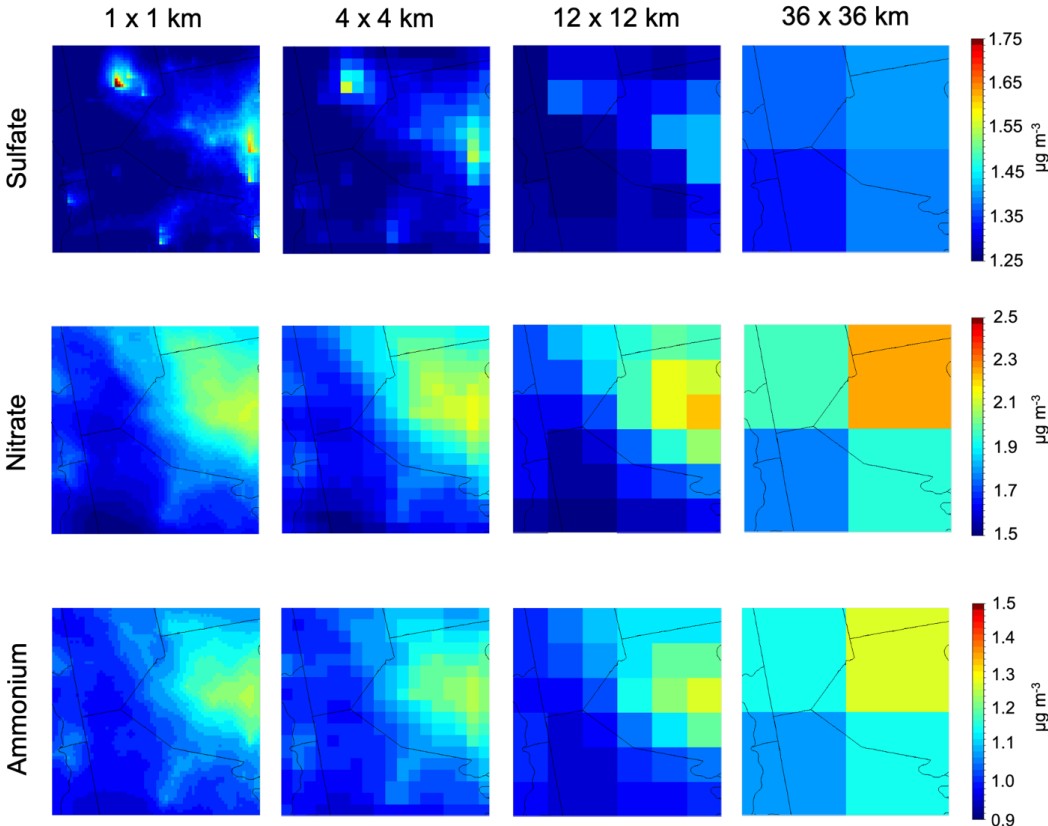


**Figure 4.** Average predicted ground-level concentration of PM$_{2.5}$ sulfate, nitrate and ammonium at a 36 x 36, 12 x 12, 4 x 4 and 1 x 1 km resolution during February 2017. Different scales are used for the various maps.








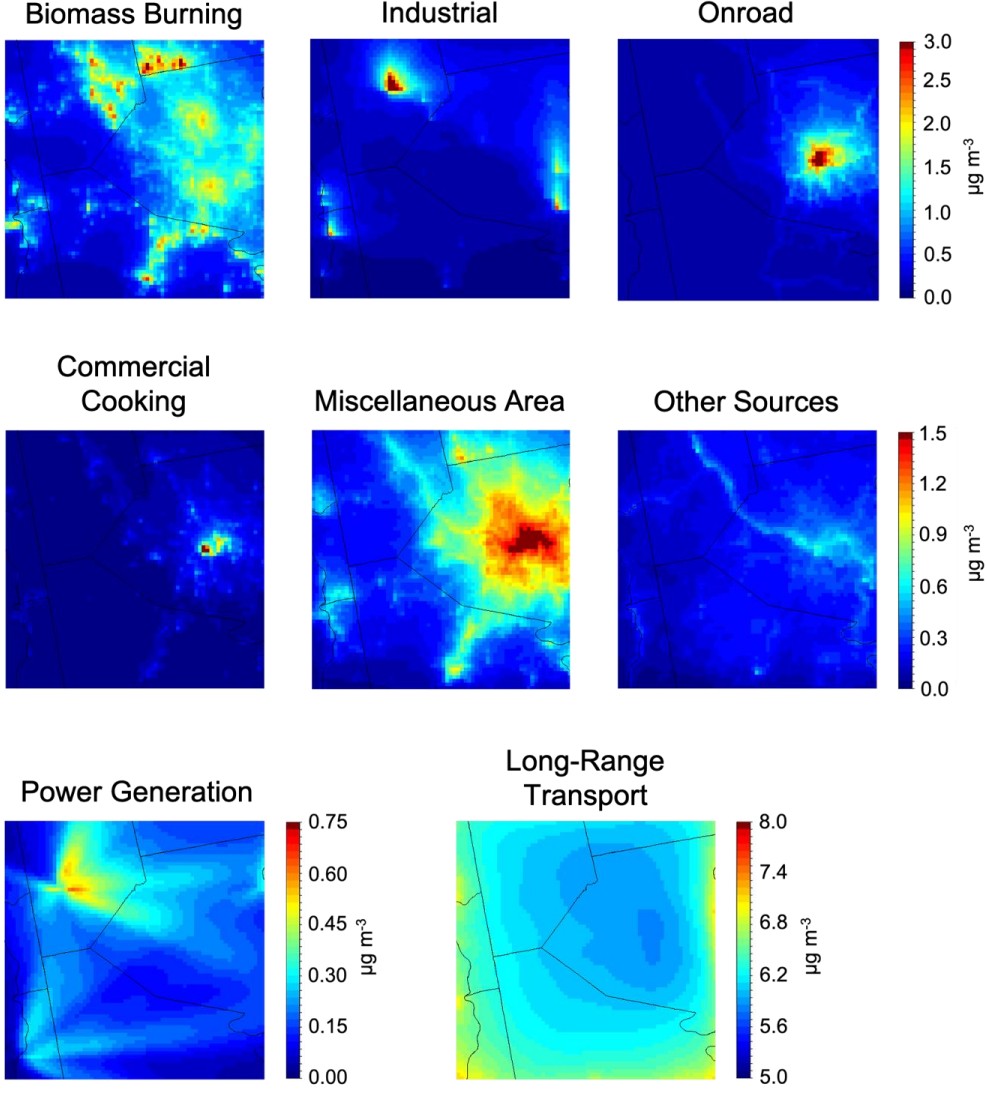



**Figure 5.** Contribution of each source to total PM$_{2.5}$ during February 2017. Different scales
are used for the various maps.




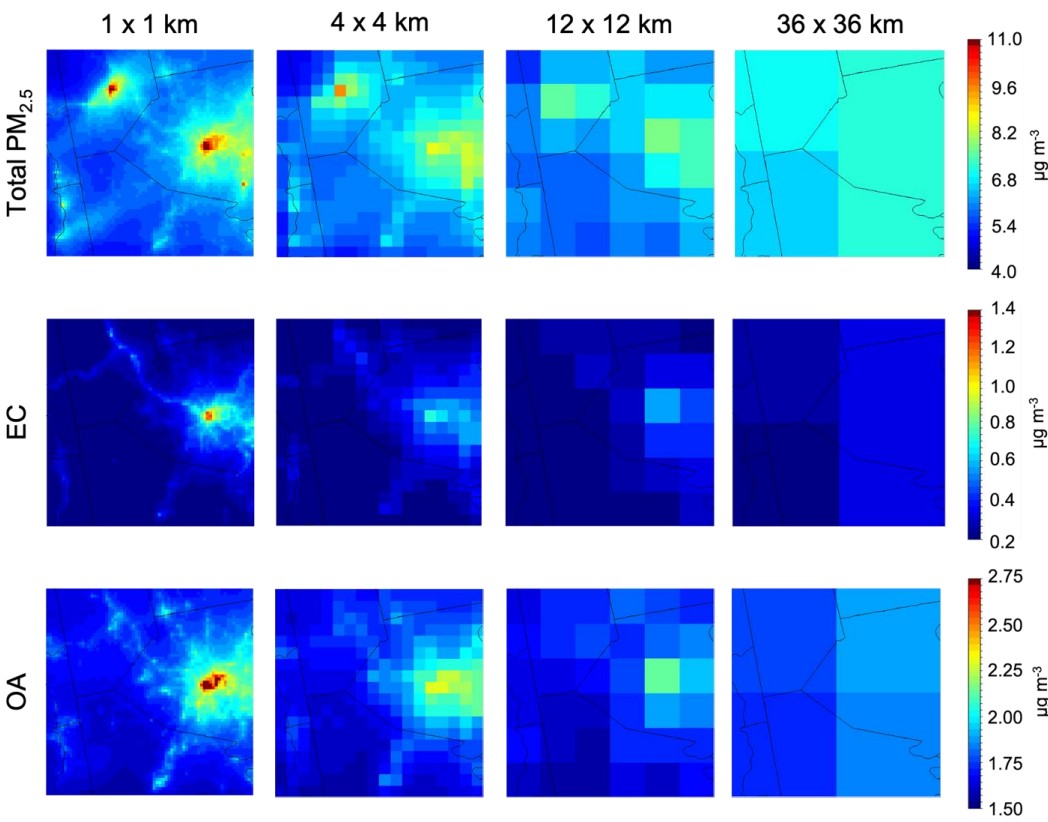

**Figure 6** Average predicted concentration at the ground level of total PM$_{2.5}$, EC and OA at a 36x36, 12x12, 4x4 and 1x1 km during July 2017. Different scales are used for the various maps.




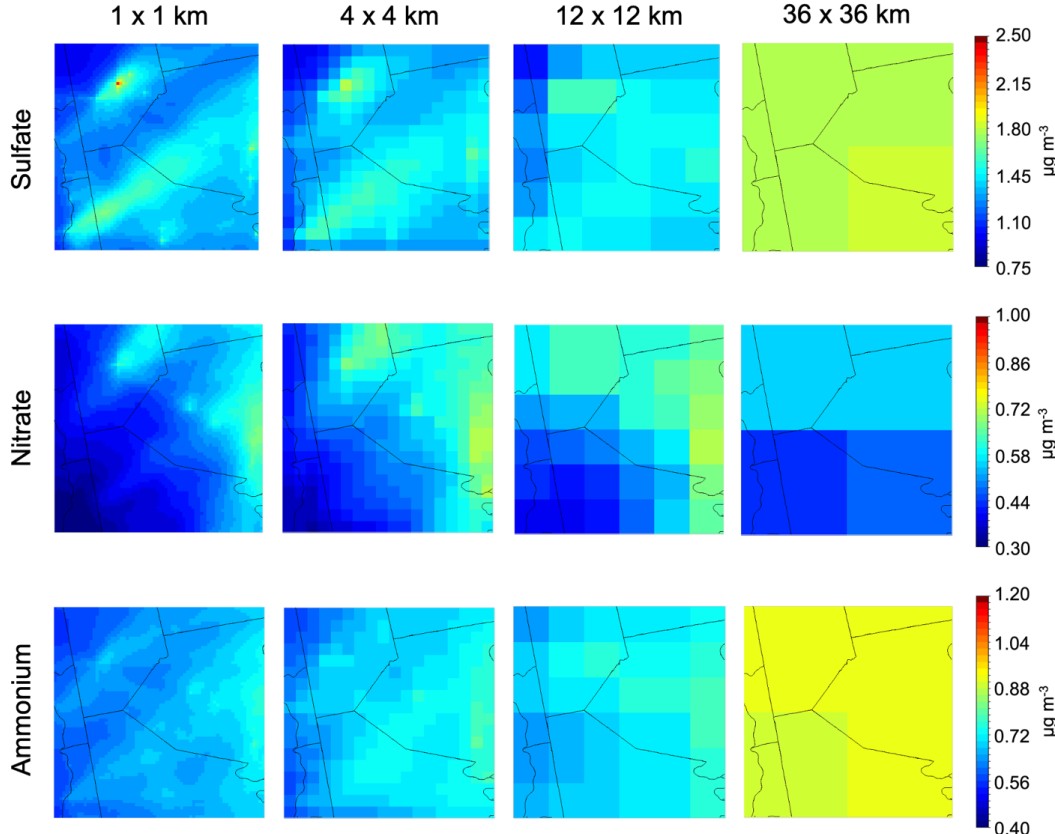


**Figure 7** Average predicted concentration of PM$_{2.5}$ sulfate, nitrate, and ammonium at a
36x36, 12x 12, 4x4 and 1x1 km during July 2017. Different scales are used for the various
maps.






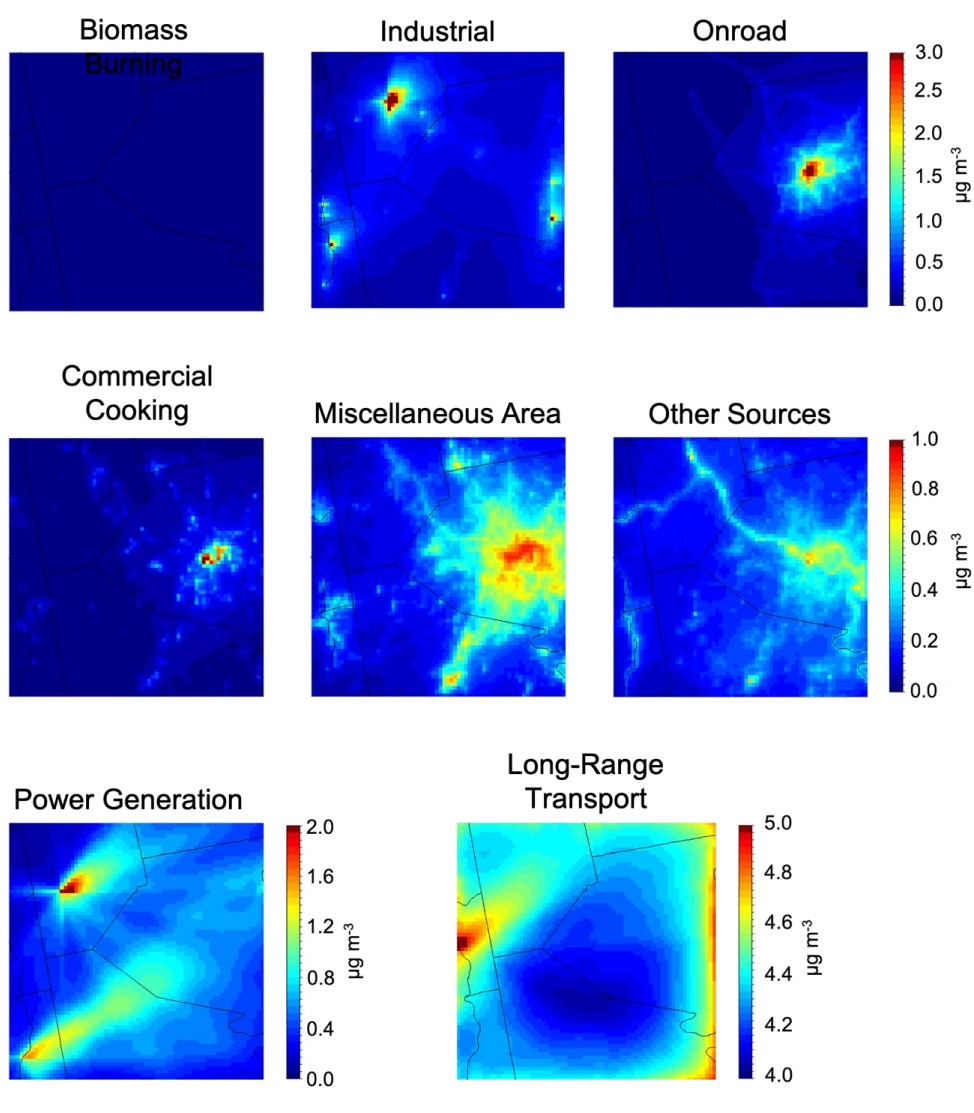



**Figure 8** Contribution of each source to total $PM_{2.5}$ during July 2017. Different scales are

used for the various maps.






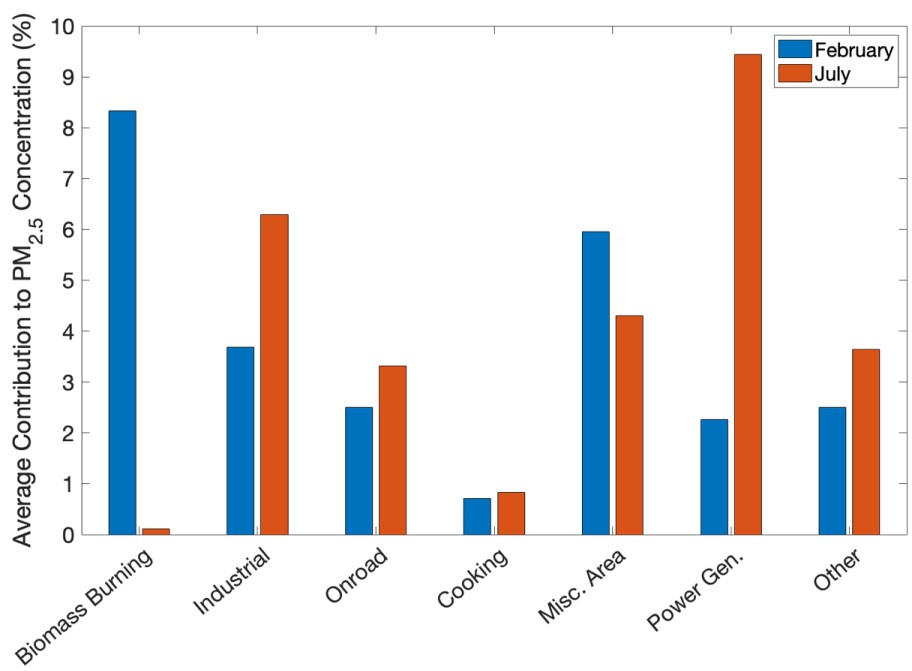



**Figure 9** Relative contributions of local sources to average predicted total PM$_{2.5}$
concentrations in the Allegheny County simulation domain during February and July 2017.




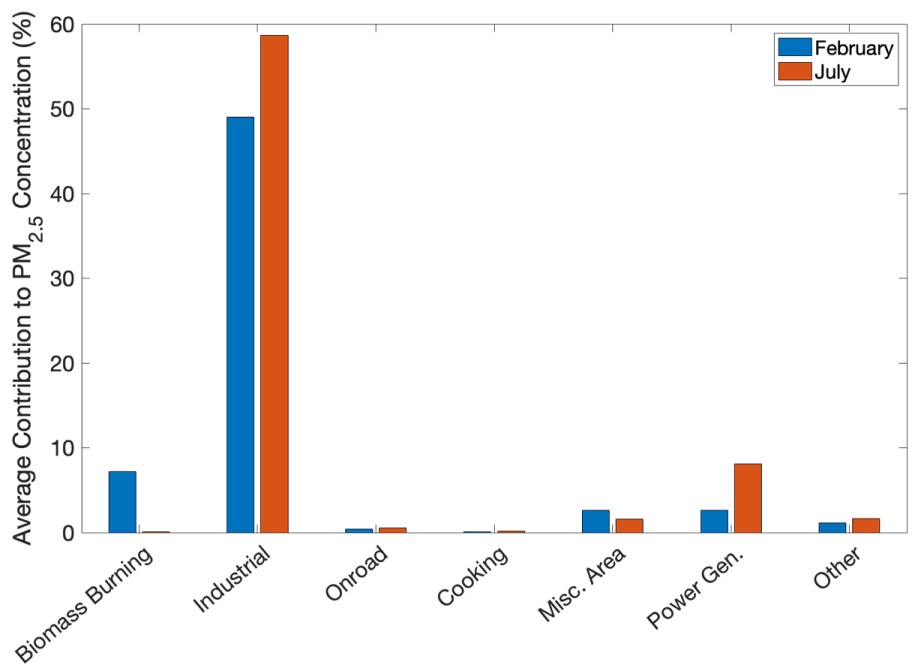


**Figure 10** Relative contributions of local sources to average predicted $PM_{2.5}$ concentrations
at the location of highest average concentration (Butler County) during February and July
2017.









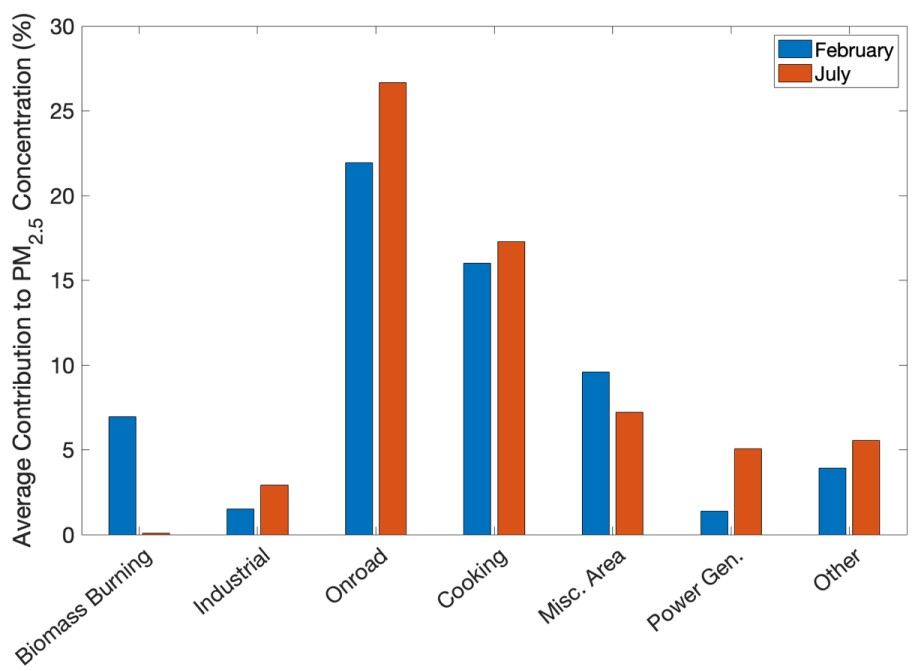


**Figure 11** Relative contributions of local sources to average predicted total PM$_{2.5}$ concentrations in downtown Pittsburgh during February and July 2017.







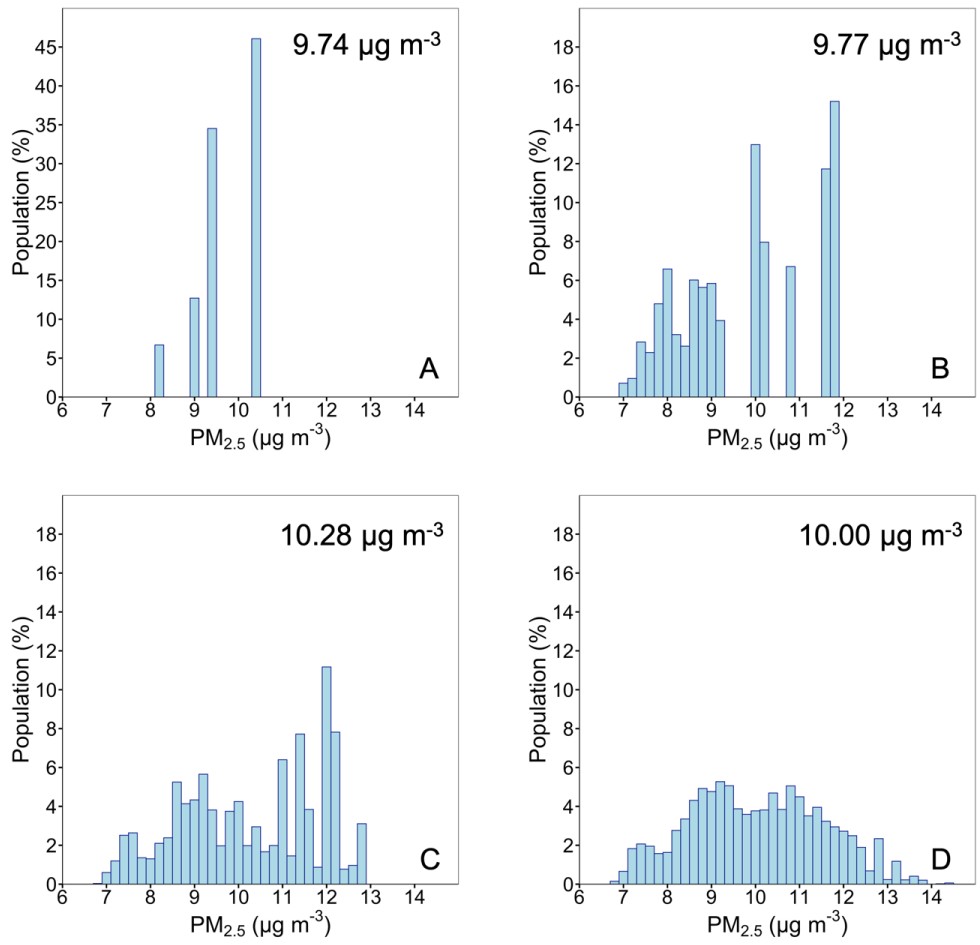



**Figure 12** Population exposure histograms at (**A**) 36x36, (**B**) 12x 12, (**C**) 4x4 and (**D**) 1x1 km during February 2017. A different scale for population is used for the distribution at 36 x 36 km resolution. The average population weighted $PM_{2.5}$ concentration for each resolution is shown in the upper right corner of each window.




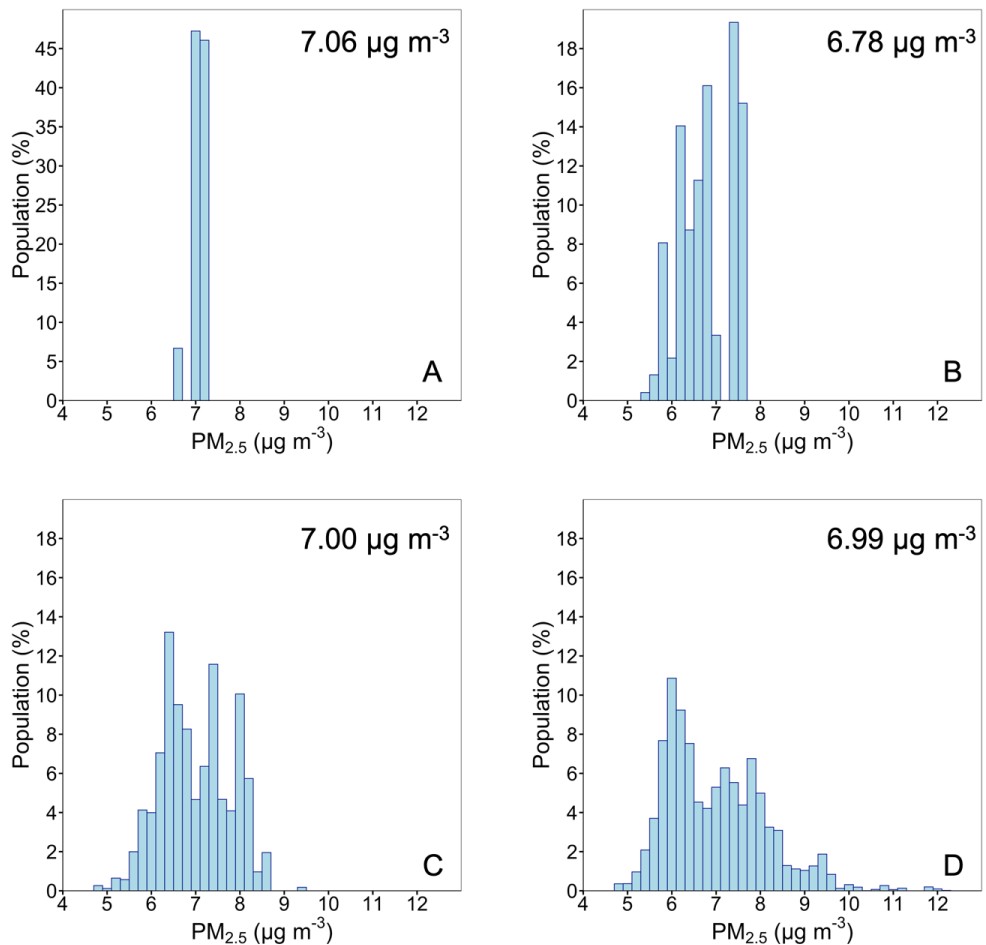



**Figure 13** Population exposure histograms at (**A**) 36x36, (**B**) 12x 12, (**C**) 4x4 and (**D**) 1x1
km during July 2017. A different scale for population is used for the distribution at 36 x 36
km resolution. The average population weighted $PM_{2.5}$ concentration for each resolution is
shown in the upper right corner of each window.







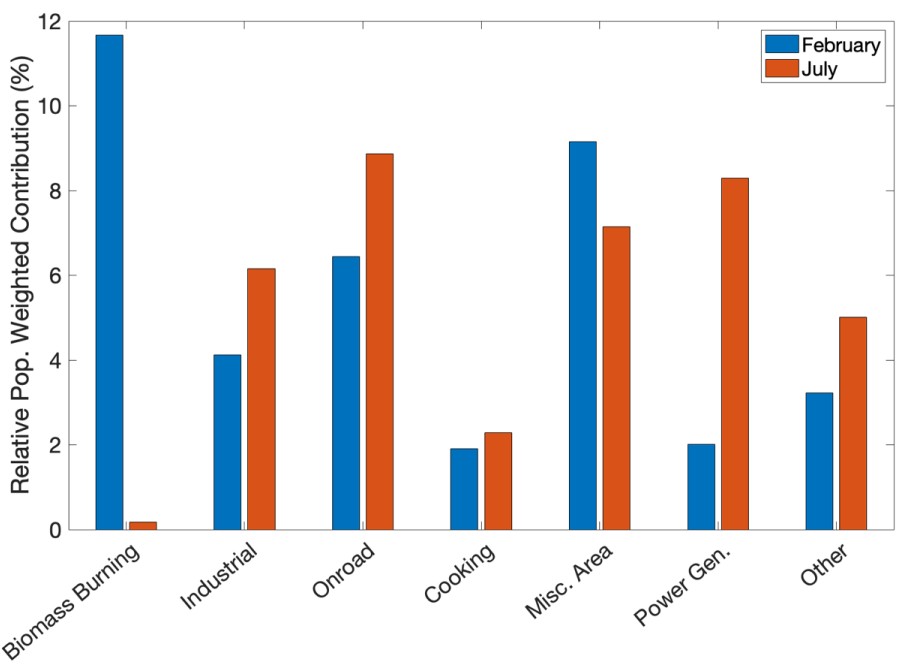


**Figure 14** Relative contributions from local sources to population weighted total PM$_{2.5}$

concentration for February and July 2017.
