# Peer review of "Source-Resolved Variability of Fine Particulate Matter and"

_Atmospheric Chemistry and Physics, 2021_

## Author Response (AR1)

**Responses to the Comments of Referee #1**

**Major comments**

(**1**) The authors note that one often limiting factor in higher-resolution modeling is the spatial allocation of the emissions data. Here they use standard approaches for most emissions sectors but use alternative methods for on-road vehicles and commercial cooking establishments. However, no evaluation of these alternative methods is provided. If the authors are reserving a full evaluation against observations for another manuscript, they could still perform a comparison of results using their new methods vs those obtained using standard approaches, or at least compare the emissions obtained using the standard and new methods.

We have followed the suggestion of the reviewer and added a comparison of the emission fields using the alternative approaches together with the corresponding discussion. The changes are especially important for the cooking organic aerosol emissions. A detailed evaluation of the model predictions against observations is the topic of a subsequent paper.

(**2**) Somewhere, either in the methods or conclusions, the authors should acknowledge that their approach of interpolating meteorological data for the 4 km and 1 km domains from the 12 km WRF simulation (rather than conducting separate higher-resolution meteorological simulations) may affect their results and limit the benefit of higher-resolution PMCAMx simulations. The same may be said for the emissions, if they are using surrogates from the 12 km domain.

We agree that the interpolation of the meteorological fields from the 12 km WRF simulation is a potential limitation. This information has been added to the conclusions as an area of future improvement. The emission fields were not based on interpolation. Data sources from which the 12 km EPA surrogates were built were used to build new surrogates at 4 km and 1 km resolutions. This information about the actual resolution of the emissions at the higher resolutions has been added to the revised paper.

(**3**) 93-112: Many of the papers that are cited in the manuscript are missing from the list of References, particularly in the model description section.

The missing references have been added to the reference list.

(**4**) 122-132: How many vertical layers were used in the WRF modeling? Were simulation results used to provide chemical lateral boundary conditions for the nested domains? I would assume that is the case, but the authors should say so.

28 layers were used in the WRF modeling. 14 of them corresponded to the vertical layers in the CTM to avoid interpolation errors. The results from the parent simulations were indeed used as chemical lateral boundary conditions for the nested grids. This information has been clarified in the manuscript.

(**5**) Tables 1 and 2: Does biomass burning include wildfires, or only residential wood combustion?

The biomass burning emissions shown in these two tables only include residential wood combustion emissions. There were no significant wildfires in this smaller domain during the simulation periods. This has been clarified in the revised manuscript.

**(6)** 142-143: The previous paragraph said custom surrogates were developed for commercial cooking for "the higher resolution grids", but this sentence suggests that the normalized restaurant count approach was used only for the 1 x 1 km grid. Please clarify.
The actual restaurant locations were used for both the 1x1 km and 4x4 km grids. This information has been added to the paper.

**(7)** 148: The emissions units $\text{kg d}^{-1} \text{ km}^{-2}$ specified for Tables 1 and 2 make sense. However, I am unable to interpret what is meant by a unit of $\text{kg g}^{-1} \text{ km}^{-2}$. Is this a typo?
This typo has been corrected in the manuscript.

**(8)** 149-153: Per Table 1, I calculate that on-road emissions are about 2.4% of total emissions. (Total emissions = 7918.5 $\text{kg d}^{-1} \text{ km}^{-2}$ for winter; 188/7918.5 = 2.4%). (Incidentally, the caption to Figure 2 should specify whether the plots are for February or July.) If emissions were uniformly distributed in space, then Fig. 2b would show 2.4% everywhere. The authors state that on-road emissions are concentrated in downtown Pittsburgh, but the on-road fraction of total emissions in Fig. 2b has a maximum value of about 1.2%. I suppose this could be because other emissions sources are even more concentrated downtown, bringing down the on-road fraction, but still it seems counterintuitive. What sector is even more disproportionately located downtown (and thus a better tracer for primary PM) than on-road emissions?
The percentage illustrated in Figure 2 refers only to the total emissions in the modeling domain by this specific emissions sector and not to the total emissions in this grid cell. In other words, the max of 1.2% is the percentage of total on-road emissions in the domain, not the percentage of total emissions from all sectors in that individual computational cell. We have improved the label of the color bar and also the figure caption to avoid this misunderstanding. The name of the corresponding month is also included in the figure caption.

**(9)** Figure 1: The maps are almost impossible to read, particularly Fig. 1B. The county(?) lines are much too faint. Perhaps it would help if the grid lines for the 1 x 1 km boxes were omitted.
The maps have been improved, removing grid lines.

**(10)** 230-234: I was quite confused trying to reconcile the figures with the text, and eventually pulled up maps from Google and Wikipedia. Butler County is to the north of Pittsburgh; the town of Butler is due north and slightly to the east of downtown, and actually outside the modeling domain. I believe that with the exception of line 230, every place in the text that says Butler (and there are MANY such places) should actually be Beaver, which indeed is northwest of downtown Pittsburgh. Making better maps would help.

We thank the reviewer for correcting this rather embarrassing geographical mistake for authors living in western Pennsylvania. We have replaced "Butler" with "Beaver" county in the manuscript.

**(11)** 245-250, 345-347: The large contribution to $PM_{2.5}$ from commercial cooking downtown (16%) is remarkable and strains credulity. At minimum, further analysis is warranted comparing these emissions to the inventory. If commercial cooking emissions are really that large, then what about residential cooking? Is this accounted for in "other sources"? Under-represented in the NEI?

Cooking OA is predicted to be 16% of the total $PM_{2.5}$ in this restaurant-dense downtown area. This rather surprising result is consistent with the measurements of Ye et al. (2018) using an AMS inside a mobile laboratory moving around Pittsburgh. These authors concluded that in the downtown Pittsburgh area, cooking OA contributes up to 60% of the non-refractory $PM_1$ mass Additionally, mobile AMS results from Gu et al. (2018) showed that cooking OA contributes 5-20% of $PM_1$ mass over a lot of areas of Pittsburgh. Even if the PMCAMx average predictions cannot be compared directly with these results, they are quite consistent with these measurements regarding the local importance of cooking OA. Similar measurements in Pittsburgh showed that the cooking OA concentrations were clearly elevated in the vicinity of restaurants in contrast to the residential areas (Robinson et al., 2018). A brief discussion of this issue together with the corresponding references has been added to the revised paper.

**(12)** 351-360: Why is the Mitchell plant plume not visible in the winter? Did the plant operate?

The Mitchell plant plume is visible in the lower left corner of the Power Generation map in Figure 5. Its plume is not as clearly defined during the winter as during the summer in the maps, because they show the ground level $PM_{2.5}$ concentrations. The emissions stack of this plant is very tall (almost 400 m) and adding the plume rise the effective emission altitude is even higher. As a result, a significant fraction of the emissions from this source is trapped above the shallow mixing layer especially during the nighttime during this wintertime period and does not reach the ground until it has been diluted. A short discussion of this has been added to the main text and a figure has been added to the supplementary material that shows the average $PM_{2.5}$ concentration from power generation in the upper air layers.

**(13)** 361-365 and Figure 8: There appears to be a concentrated plume at the central portion of the western boundary of the modeling domain. Is this a wildfire?

This high concentration area is indeed to the transport of $PM_{2.5}$ from outside the inner domain. It is actually due to power plants and other industrial sources in the Ohio River valley and not to a wildfire. This point has been added to the revised paper.

**(14)** 392: What is the resolution of the population data? Is it available at 1 km resolution? Given that one of the principal conclusions of this paper concerns population-weighted PM concentrations, more discussion of the population data is warranted.

Population data is at the census block group level which is smaller than our grid cell size. This has been clarified in the manuscript.

**(15)** 430: This should refer to Figure 13 (or perhaps the figures should be reordered).
This typo has been corrected in the manuscript.

**(16)** Figure 8: Fix the "Biomass Burning" caption so that it is one line.
This has been fixed in the manuscript.

**(17)** Figure 9: This caption refers to the "Allegheny County simulation domain", which is not mentioned elsewhere in the text. Should this just say "downtown Pittsburgh"?
This refers to the entire inner domain. References like this have been adjusted in the manuscript for consistency.

**Responses to the Comments of Referee #2**

**Major Concerns**

(**1**) Figure 9 provides a helpful framing of the source-oriented contributions to total $PM_{2.5}$. I am concerned about its interpretation though considering Tables 1 and 2. For example, power generation in Table 1 is nearly double the $PM_{2.5}$ emissions of biomass burning. But in Fig. 9, it is about one third of the biomass burning contribution for February. Is this an issue of biomass burning SOA contributing heavily to $PM_{2.5}$ in winter, or has the massive 'other' PM category for power generation potentially been thrown away? In July, Table 2 suggests the power generation contribution to $PM_{2.5}$ should be more like 60%, not 9.5%. This issue is critical also for understanding and discussing Fig. 10. Please confirm that power generation is not seriously underrepresented here.

These are all good points that deserve additional discussion in the paper. The small contribution of power generation compared to biomass burning in the winter period is largely due to the vertical distribution of the corresponding aerosol in a period with relatively low mixing heights. PMCAMx predicts that a large fraction of the power generation emissions is emitted aloft and stays aloft for a considerable period without affecting the ground concentrations in the nearby areas. This is contrast to the residential biomass burning emissions that are mostly below the mixing height and therefore are rapidly mixed down to the ground level. Discussion of the upper air concentrations of biomass burning and power generation $PM_{2.5}$ as well as the "other" emissions category has been added to the main text. A figure showing the average upper air concentrations of biomass burning and power generation $PM_{2.5}$ in the winter has been added to the supplementary material.

For the July period, one needs to take into account that Table 2 is showing local emissions (in the Pittsburgh 1x1 km inner domain) while Figure 9 shows the contribution of these local sources to the total $PM_{2.5}$ mass. A significant fraction of the $PM_{2.5}$ is not local (it is transported from other areas) and therefore the contribution of the local sources is much lower than what Table 2 suggests. As a result, the 9.5% refers to the local power generation. Obviously, the contribution of all power generation sources is much higher than this. We have explained this important point in the revised paper to avoid confusion.

(**2**) I don't believe the city northwest of Pittsburgh is Butler – it's Beaver. Note that the county directly northwest of Allegheny is Beaver County. Please update this throughout the text and figures. It looks like the particle emissions are mostly sulfate in Fig. 4. If these emissions are due to one source, it would be interesting to identify it. Other well-known sources like the Clairton Cokeworks receive public attention in Allegheny county for their proximity upwind of downtown and yet this Beaver County source (or sources) appears to be crucial for understanding and managing air quality there (160,000+ people).

That is correct. Most of these references to Butler should actually be Beaver. The corresponding references have been changed in the paper.

**(3)** The authors have overlooked a dramatic result from this high-resolution exercise – according to the model, the residents of Beaver, PA are exposed to similar or even higher PM$_{2.5}$ concentrations as if they lived directly in downtown Pittsburgh. I'm sure this would come as a surprise to most of them (especially since it's not EC and thus less routinely visible) and is not captured well by the standard 36 km or 12 km models. Thus reduced-complexity tools like EASIUR, which I believe is resolved at 36 km, likely miss it and managers have potentially underestimated it as well. And to the extent that PM$_{2.5}$ from power generation (see major point 1) may be underestimated by the model, the results may be even more concerning than shown here. The authors provide a couple of figures and basic discussion of the results in Beaver, but I think this dataset provides a real opportunity to frame the analysis from the perspective of EJ, since a relatively small number of people are impacted by a few important point sources that could be regulated with more ease than distributed sources like residential wood or volatile chemical products, for example. In recent years, Shell has opened a massive ethane cracker facility located exactly within the sulfate emission hot-spot, so it would be interesting to discuss the potential impact of that new source, if the authors can find basic annual emission data for that facility (and its support operations) to put it in context with the results presented here.

This is a good point; a discussion has been added to the main text to highlight this. This is illustrated in Figures 12 and 13. These people who experience the highest concentrations in the modeling domain (Beaver County) do not really show up in the estimated exposure distribution until the model reaches the 1x1 km resolution. Indeed, this is more apparent in the July period, where at 1x1 km resolution a larger range (8.5 – 12 µg m$^{-3}$) of concentrations is predicted. While the range of concentrations in the upper tail of the exposure distribution from the winter simulations is a little smaller, this is again partially due to the vertical distribution of the aerosol in the winter.

**(4)** Figures 12-14 illustrate an interesting issue for exposure assessments. We typically think of population-weighting as being the best way to translate air quality model fields to exposure estimation. But in the context of EJ, the real impacts on communities in the upper tail of exposure are obfuscated by a population-weighted approach. I would love to see the authors use the data in this manuscript to clearly amplify this interesting point.

This is a good point and a brief discussion of its implications has been added in the revised paper. We do plan to extend this work to EJ issues and address this issue in more detail.

**(5)** It is unclear to me how the fact that population-weighted average concentration is similar among the different resolution cases necessarily means that higher resolution data are not useful for epidemiological work. Is this statement based on the assumption that health data would be matched at the county-level? If census-tract data were used, would the authors' conclusion be different?

This is also an important point. We now clarify that this statement is based on the assumption that the available health data for the epidemiological analysis are at the county level. If the health data

of interest is at the census tract level, the high-resolution PM$_{2.5}$ and the calculation of more accurate exposure would be important for the epidemiological study. This has been clarified in the revised manuscript.

**Minor Suggestions/ Typos**

**(6)** Line 31: Consider adding 'primary' before organic aerosol.
Added.

**(7)** Line 43: 'reduced lung development and function in children, reduced function in people with lung diseases such as asthma, and…'
Corrected.

**(8)** Lines 102-104: For fine and coarse PM, or just for fine?
Both fine and coarse PM are included in the model and are predicted. The analysis in this paper focuses on PM$_{2.5}$. This has been clarified in the manuscript.

**(9)** Line 104: partitioning
This has been fixed in the manuscript.

**(10)** Line 105: What version of ISORROPIA?
ISORROPIA-I (Nenes et al., 1998) was used for inorganic aerosol thermodynamics. This has been clarified in the manuscript.

**(11)** Line 106: Can you describe the volatility distributions used to describe the POA from the various sources? And what assumptions are being used for IVOC emissions across sources? A lot of information has been written about this over the years, so no need for long discussion, but perhaps a table summarizing the parameter choices (in the SI?) would be helpful. This is particularly important for comparing sources of OA emissions in Tables 1 and 2 and then interpreting Fig. 9.
The volatility distribution for POA by Tsimpidi et al. (2010) was used in these simulations for all sources. This information together with a brief discussion has been added to the paper.

**(12)** Line 109: second aSOA should be bSOA.
Corrected.

**(13)** Section 2: Can you add some details about the aqueous-phase chemistry package and version?
Aqueous-phase chemistry is simulated using the Variable Size Resolution Model (VSRM) of Fahey and Pandis (2001). This reference has been added to the manuscript.

**(14)** Lines 115-120: How much spin-up?

Two spin up days were thrown out at the start of each simulation. This has been clarified in the manuscript.

**(15)** Line 120: How long does it take the model to run each domain?

Simulations took around 6 days, 5 hours, 10 hours, and 12 days for the 36 km, 12 km, 4 km, and 1 km domains, respectively. This has been clarified in the manuscript.

**(16)** Line 122: Table S1 does not have this information.

This problem has been fixed. Table S1 had been placed in the file with the main text instead of the Supplementary Information file.

**(17)** Lines 126-127: Interpolating the met fields down is risky and might break the density continuity equation in a big way. Are you concerned about the impact this might have in a place like Southwestern PA with relatively extreme elevation variability, especially in populated areas (i.e. valleys)? An easy way to address this would be to provide some basic evaluation results (e.g. $O_3$, $SO_2$, $NO_2$, total PM, etc.), although the I understand you want to save the bulk of that discussion for a follow-on paper.

The resolution of the meteorology does introduce some uncertainty in our predictions. This issue is addressed in some detail in the subsequent evaluation paper. Unfortunately, in some industrial areas with narrow valleys even the 1x1 km resolution for the meteorology is probably not sufficient for the description of the dispersion of the corresponding plumes. This is now noted in the paper.

**(18)** Tables 1 and 2: What is the "other" material coming from power generation? Presumably there are lots of metals in here, but what else? Are these dry particles? Is this mass included in an 'Other' category in PMCAMx or is it neglected?

The power generation emissions described in Tables 1 and 2 refer to dry particle mass. A lot of the 'other' category is ash including the corresponding metals. These are simulated by PMCAMx as inert particle mass. This has been clarified in the main text.

**(19)** Can you also explain in the main text whether these emissions are reflective of filterable or condensable PM for the power generation sector?

According to the NEI Technical Support Document, all $PM_{2.5}$ contains both condensable and filterable particulate matter (U.S. Environmental Protection Agency, 2015). This has been added to the paper.

**(20)** Lines 133-141: Were coarse-mode PM emissions considered?

Both fine and coarse emissions were included and the full size-composition distribution has been simulated by PMCAMx. Our discussion in the paper has focused on the fine PM results. This has been clarified in the manuscript.

**(21)** Lines 142-153: Were the cooking and traffic emissions calculated specifically for 2017, projected, or estimated for another similar year and pasted in?

These were also projected emissions for 2017. This is mentioned at this point.

**(22)** Figs 3, 4, 6, and 7: It took me a while to figure out that these maps have a variable lower color limit. I think that gives the false impression that there are no emissions in areas of the domain for certain species or sources. For example, Fig. 5 shows power generation is elevated in Beaver county, but these emissions are not pictured in Figs. 3 or 4. They should show up in the sulfate map, no? I recommend adjusting the speciation maps to using log color scales and reducing the lower limit to 0. Or you could perhaps choose some clever discontinuous ticks for the axis but that would be tough considering the purpose of showing continuous changes due to increased resolution moving right to left in the panels.

The baselines have been chosen to effectively remove the background from these plots and highlight the effect of local sources. The power generation emissions do show up in Figure 3, although the scale does make it a bit difficult to see, compared to the plot where all contributions other than local power generation have been removed (Figure 5). The sulfate maps appear to be the main culprit here. The scale for these plots has been fixed to make this clear. The varied lower limit has also been highlighted by a comment in the main text as well as in the figure captions.

**(23)** Section 4.1: All components from figures are discussed except sulfate. It would be interesting to discuss, especially in the context of the other inorganics and spatial refinement of aerosol pH predictions.

The corresponding discussion of sulfate has been added to the manuscript, including the implications of power generation plume resolution during this simulation period.

**(24)** Lines 255-257: Probably worth mentioning that the Bruce Mansfield power plant was shut down in 2019.

This comment has been added to the manuscript.

**(25)** 8: Probably want to move the 'Biomass Burning' label completely outside of the map.

This has been fixed in the manuscript.

**(26)** Lines 381-388: Fig. 11 is referenced but those results are not discussed.

A short discussion on this has been added to the manuscript. The main result from this figure are that the majority of PM$_{2.5}$ emissions in the downtown area can be attributed to either traffic or cooking, in both simulation periods.

**(27)** Lines 413-416: This statement is fundamentally neutral, but the authors may want to rephrase considering they have tied this work to the goals of Environmental Justice. Some could interpret

this statement to suggest that the population density being low indicates the problem is not meaningful. I recommend tying this result directly to Environmental Justice (see major point 3). The same is true for the summer period, where the large values in the 1x1 km case (> 11 ug m-3) are not even mentioned in the discussion.

Indeed, this has been rephrased to highlight the Environmental Justice perspective of this result.